# Management of Femur Fractures during COVID-19 Pandemic Period: The Influence of Vaccination and Nosocomial COVID-19 Infection

**DOI:** 10.3390/jcm11226605

**Published:** 2022-11-08

**Authors:** Marianna Faggiani, Salvatore Risitano, Alessandro Aprato, Luigi Conforti, Alessandro Massè

**Affiliations:** 1ASL TORINO 5, Department of Orthopaedic Surgery and Traumatology, 10024 Turin, Italy; 2Città della Salute e della Scienza di Torino, Department of Orthopaedic Surgery and Traumatology, 10126 Turin, Italy

**Keywords:** COVID-19, femur fractures, COVID-19 vaccination

## Abstract

The COVID-19 pandemic management has led to a significant change in orthopedic surgical activity. During the pandemic, femur fractures in patients over 65 years of age have maintained a constant incidence. Our study will focus on this fragile population, analyzing the incidence of SARS-CoV-2 infection during hospital stays and the clinical and radiographic orthopedic outcomes. We also evaluated the va\riation of COVID-19 infection after health professionals’ vaccinations, and the influence of inter-hospital transfers caused by logistical and organizational aspects of the pandemic. **Material and Methods:** This is a descriptive and prospective study from 13 October 2020 to 15 March 2021. Participants were patients over 65 years of age with diagnoses of proximal femoral fractures with r surgical treatments indicated. We compared the SARS-CoV-2 infected patients during the stay with non-infected cases. A second evaluation was carried out dividing the patients into those who underwent inter-hospital transfers and a group without transfers. We subdivided the study period into two, according to the percentage of healthcare workers vaccinated against SARS-CoV-2. The reported clinical variables included the Parker and Palmer Score, the Nottingham Hip Fracture Score, the Harris Hip Score, mortality, the Rush Score, and evaluation of reduction in radio-lucent lines in prosthetic implants. **Results**: Ninety-three patients were studied. The whole positive COVID cohort (11.83%) was hospitalized during the period when less than 80% of health workers were vaccinated (*p* = 0.02). The COVID cohort and the patients transferred before surgery had longer stays in the Emergency Room (*p* = 0.019; *p* = 0.00007) and longer lengths of stay compared to the other patients (*p* = 0.00001; *p* = 0.001). Mortality was higher both in the infected group and in the patients who underwent a transfer before the surgical procedure (18.18% vs. 1.22 %; *p* = 0.003. 25% vs. 6.85%; *p* = 0.02). In terms of orthopedic outcomes measured through the third month of follow-up, we found worse score results in functional and radiographic outcomes in the COVID positive cohort and in the transferred patients’ cohort. **Conclusions**: The impact of the COVID-19 pandemic on patients treated for proximal femur fracture was statistically significant. Patients with Coronavirus during hospitalization obtained poor short-term radiographic and functional results and increased peri-operative mortality. The incidence of intra-hospital infection was high during the period in which health professionals were not yet covered by the anti-COVID vaccination cycle. Patients who were transferred between two hospitals due to pandemic-related management issues also achieved reduced outcomes compared to non-transferred cases, with increased mortality.

## 1. Introduction

Coronaviridae is a family of viruses with a single-stranded RNA genome. SARS (Severe Acute Respiratory Syndrome) is an atypical form of pneumonia caused by the Coronovirus-1 [1,2]. This disease produced an epidemic in China that developed from November 2002 to July 2003. During autumn of 2019, the health authorities of the city of Wuhan (China) found the first case of a patient showing a different respiratory disease, referred to as “pneumonia of unknown cause” [3,4]. The cause was subsequently identified as a new type of virus classified as Coronavirus-2 (SARS-CoV-2) [2,5]. This virus spreads through respiratory droplets and aerosols produced by the infected subjects. It exhibits an initial nonspecific symptomatology like flu with cough, fever, and dyspnea. The condition can evolve into severe hypoxic respiratory failure [6,7].

The World Health Organization (WHO) declared a SARS-CoV-2 pandemic on 11 March 2020 [6]. Italy was the first European nation to face this health emergency. Northern Italy was more involved than the rest of the country [8]. Measures imposed for contagion containment upset every aspect of society and subverted hospital organization, altering the incidence of traumatic pathology [9,10,11]. During the first pandemic period, road accidents were reduced by 77% and sports accidents by almost 100% [9,12]. Accidents at home experienced a minor increase [5]. A systematic review carried out by the Orthopedic Surgeons of Wuhan (China) showed that the incidence of SARS-CoV-2 infection in orthopedic wards was almost 20% more than the incidence among total inpatients [4,13].

Guidelines (16 March 2020) issued by the Italian Society of Orthopedic and Traumatology (SIOT) indicated that orthopedic and traumatological surgery cannot be suspended and must be reorganized instead [3,4]. During the pandemic period, femur fractures in patients over 65 years old maintained a constant incidence [14,15]. These elderly fractures remain a surgical priority [16,17]. These fragile patients need to walk as early as possible, and be allowed rapid rehabilitations and reduced hospitalization time [18,19]. The literature has shown that early surgery leads to a significant reduction in mortality and peri-operative complications such as urinary tract infections (2.5%), respiratory complications (4.5%) and cardiac (3.2%) or decubitus injuries (2.4%) [20,21,22,23]. Moreover, according to more up-to-date studies, a concomitant infection by SARS-CoV-2 leads to an increase in complications and perioperative mortality in these surgical orthopedic patients [17,18]. In a multicentric study, 89% of positive patients who presented post-operative complications greater than the negative and 20% who experienced respiratory distress syndrome and multiorgan insufficiency [18,19]. On 13 October 2020, the Italian Infective Disease Department prolonged the emergency period. On 21 December 2020, the European Medicine Agency (EMA) authorized the first vaccine against SARS-CoV-2, called COMIRNATY (developed and produced by Pfizer/Biomtech). The Italian Drug Agency (AIFA) approved COMIRNATY the next day; therefore, the vaccination campaign against SARS-CoV-2 was launched on 27 December. The national strategic plan provided for vaccination first of health staff and fragile guests of the Health Care Residences. The World Health Organization recommended that individual governments identify vaccine hesitancy areas [9,10]. In Italy, health workers who opposed vaccination were suspended.

Our study will focus on patients over 65 years old with proximal femur fractures, analyzing the incidence of the inpatients’ onset of SARS-CoV-2 infection and its negative influence on clinical and radiographic orthopedic outcomes. We also will analyze variations in the incidence of SARS-CoV-2 infections among patients after the health professionals were vaccinated and the influence of inter-hospital transfers (caused by pandemic related logistical and organizational issues) in this fragile population.

## 2. Material and Methods

### 2.1. Study Design and Participants

This is a descriptive and prospective study from 13 October 2020 (on the day that the Italian government prolonged the state of national alarm due to COVID-19) until 15 March 2021 [6,8,9]. Included participants were patients over 65 years of age presenting to our Emergency Department with clinical and radiographic diagnoses of proximal femoral fractures (31-A-B and C according to the OTA/AO classification) with indications for surgical treatment. Exclusion criteria were patients with femoral shaft fractures, open fractures, pathological fractures, periprosthetic or peri-implant fractures, polytrauma, or nonoperative fractures and patients diagnosed with COVID-19 (determined by a polymerase chain reaction, PCR, test from nose swab samples at the entrance to the Emergency Department) [24]. Our department of orthopedics and traumatology covers an area distributing the work between two different hospitals with patients present in both emergency rooms, operating rooms, and orthopedic wards. During the emergency period, according to national health restrictions (D.L. n. 125 of 7 October 2020, converted into law n. 159 of 27 November 2020), our health department planned the transfer of all surgical patients to a single reference hospital, leaving open only the services of the E.R. in the other one. The health professionals were equipped with every protective device and were subjected to anti-SARS-CoV-2 vaccination beginning 1 January 2021. All patients infected by the COVID virus during the stay were transferred to a COVID-19 ward. The elderly population is more immune compromised. They developed an inflammatory storm syndrome that further complicates the host defense mechanism [25]. For symptomatic patients a corporate protocol based on steroids, antivirals, and oxygen therapy was used.

All surgical procedures were performed with the same implant (Gamma 3 Nail Stryker for internal osteosynthesis and Gladiator Bipolar System for the arthroplasty) and by the same surgical team composed of four orthopedic specialists. The choice of cementation during the arthroplasty procedure was made at the time of surgery according to the bone stock. To compare the data, we divided the sample into two groups: patients who were SARS-CoV-2 infected during the stay, diagnosed by a PCR test from nose swab sample (Group A), and cases not infected (Group B). A second evaluation was carried out dividing the patients into a sample that underwent an inter-hospital transfer (Group C) and a group without any transfers (Group D). We subdivided the study period into two, according to the percentage of healthcare workers vaccinated against SARS-CoV-2 (with double doses of Pfizer/Biomtech): Time 0 (from 15 October 2020 to 10 February 2021), when the percentage of vaccination was less than 80% and Time 1 (from 11 February 2021 to 15 March 2021), when that percentage was more than 80%.

The main objective was to analyze the impact of surgical logistic management during the COVID-19 pandemic on fragile patients with proximal femur fractures. We focused the analysis on the clinical and radiographic orthopedic outcomes (at time of 3 months of follow-up) and the mortality incidence of patients who were infected by SARS-CoV-2 during the stay compared to patients not infected. Secondly, we wanted to evaluate the variation of SARS-CoV-2 incidence in this elderly population before and after the health professionals’ vaccinations (Time 0 vs. Time 1) [12]. Our third goal was to analyze the influence of the inter-hospital transfers on the orthopedic outcomes and mortality incidence in proximal femur fracture patients. The Institutional Review Board of our institution defined this study as exempt from IRB approval (descriptive study) and was conducted in accordance with the ethical standards laid down in the 1964 Declaration of Helsinki and informed consent to the processing of data was obtained from all patients at the entrance to the hospital.

### 2.2. Data Collection

All data were collected prospectively from the electronic medical records by only one investigator (an orthopedic resident). Demographic variables were sex, age, and residence (nursing home or family home). The reported clinical variables included the type of fracture (according to AO/OTA classification 31 A, B and C) [24], the American Society of Anesthesiologists (ASA) classification, comorbidity, pre-trauma mobility (calculated by the Parker and Palmer Score, PPM Score) [15], and risk of mortality in the 30 days post-surgery (according to Nottingham Hip Fracture Score, NHFS) [15]. The laboratory variables included hemoglobin (Hb); the number of post-treatment transfusions (our anesthesiologic protocol recommends transfusion of two bags of hematite below 10 g/dl of hemoglobin for cardiopathic patients); the type of surgical procedure performed (fracture fixation or hip replacement); the surgical procedure and physiotherapy (post-operative treatment was performed according to the same rehabilitation protocol); delay in days since presentation to the Emergency Department; oxygen therapy during the stay; number of transfers; lengthening of stay; SARS-CoV-2 related variables (PCR SARS-CoV-2 test results); abnormality of the pulmonary clinical picture radiographically evaluated; state of vaccination against SARS-CoV-2 of health workers and inpatients; type and number of post-surgical complications; range of motion and functional outcomes (expressed by the Harris Hip Score at 30 days and 3 months after surgery) [26,27]; evaluation of the antero-posterior and lateral radiographic views at 30 days and 3 months post-surgery (according to the Rush Score for the internal osteosynthesis procedures; and evaluation of the reduction of the radio-lucent line in prosthetic implants) [28].

### 2.3. Statistical Analysis

The statistical analysis of the data obtained was carried out using the software Statistical Package for Social Science version 22.0 for Macintosh (SPSS)^®^ (IBM Corp, Chicago, IL, USA). Continuous variables were presented as the mean and the standard deviation, and categorical variables were presented as the number and percentage. We used the Student’s Test T, the Mann-Whitney U test, and the chi-square test to compare differences between ordinal and categorical variables where appropriate. Statistically significant results for values of *p* < 0.05 were considered relevant. The force of the correlation identified among the continuous variables was subsequently analyzed using Spearman’s Rho and the force of the correlation among the ordinal variables was analyzed with Kendall’s Tau-b.

## 3. Results

Over the study period, 117 patients with neck femur fracture were admitted. 20.51% (*n* = 20) of cases were excluded because they did not satisfy the required criteria. At last, the total sample included 93 patients. Table 1, Table 2 and Table 3 show a summary of the main variables collected. The average age of the sample was 83.75 years (65–98, DS 19.3), 21.50% (*n* = 20) male and 78.5% (*n* = 73) female. Before the trauma, a percentage of 83.87% (*n* = 78) lived in their private home. According to the Parker and Palmer score, 6.51% (*n* = 7) of patients had a pre-trauma mobility score of less than three points, 40.92% (*n* = 44) between four and five points, and 31.62% (*n* = 34) over 5. On average, our sample reported a NHFS score of 5.24% (2.8–11.8) and mode 4.6 (DS 2.67). A percentage of 13.95% (*n* = 15) showed no significant comorbidity at the trauma time, 59.52% (*n* = 64) between one and three comorbidity and only 7.44% (*n* = 8) more than three concomitant diseases. A percentage of 23.25% (*n* = 25) had an ASA score of two40.92 (*n* = 44) of three and only 16% (*n* = 17) of four. In 3.23% (*n* = 3) was diagnosed with a femoral fracture OTA/AO 31A1, in 45.16% (*n* = 42) 31A2 and in 12.91% (*n* = 12) 31A3. Fractures type 31B/C corresponded to 31.26% (*n* = 30). A percentage of 55.91% (*n* = 52) were treated with internal synthesis, 20.43% (*n* = 19) with partial hip replacement and only two subjects (2.15%) were managed with total hip replacement.

### 3.1. COVID-19 Positive Cohort Vs. COVID-19 Negative Cohort

Among the sample, 11 (11.83%) were confirmed COVID positive by testing after the surgical procedure (Group A). Comparing demographic characteristics of Group A to Group B (COVID-19 negative cohort), the average age (*p* = 0.31), the gender (*p* = 0.41), the ASA score (*p* = 0.40), the PPM Score (*p* = 0.38), the NHFS (*p* = 1.22) and the type of fractures were comparable (*p* = 0.10) (Table 2). In terms of hospital quality measures, the whole positive COVID group was hospitalized during the period when less than 80% of health workers had been vaccinated (Time 0) (*p* = 0.02) and 72.73% (*n* = 8) needed high-flow oxygen and admission to the Intensive Care Unit (*p* = 0.000019). Group A had a longer stay in the Emergency Room (E.R.) compared to Group B (*p* = 0.019): a percentage of 63.64% (*n* = 7) of the first group remained in the E.R. more than 24 h, compared to only 22% (*n* = 18) of Group B. The positive cohort had a longer length of stay compared to the other patients (average of 21 days vs. 14 days, *p* = 0.00001). A percentage of 18.18% (*n* = 2) of infected patients, had died in the ward after the surgical procedure compared to only 1.22% (*n* = 1) of the not infected patients (*p* = 0.003). In terms of orthopedic outcomes measured to the third month of follow-up, we identified a worse score in functional (HHS 80–89 points: 18.18% vs. 33.33%) and radiographic (Rush Score 18–24 Rush Score: 10.20% vs. 40.42%) outcomes in the COVID positive cohort (*p* = 0.00001; *p* = 0.00002). SARS-CoV-2 infection during the stay and mortality relationship after discharge was also significant: 36% (*n* = 4) of subjects of Group A died in around three months after discharge compared to only 6% (*n* = 5) of the second group (*p* = 0.007).

### 3.2. Transferred Patients’ Cohort Vs. Not Transferred Patients’ Cohort

Twenty (21.51%) patients were transferred before the surgery (Group C) because of pandemic related logistics. Comparing Group C vs. D (not transferred patients), the average age (*p* = 0.61), the gender (*p* = 0.71), the ASA score (*p* = 1.40), the PPM Score (*p* = 0.22), the NHFS (*p* = 0.45) and the types of fractures were comparable (*p* = 2.10) (Table 3). Ten percent% (*n* = 2) of Group C and 24.66% (*n* = 18) of Group D underwent surgery within 24 h from the time of E.R. access (*p* = 0.00007). The surgery was delayed beyond 24 h (within 48 h) in 55% (*n* = 11) of transferred patients’ cohort than the 45.20% (*n* = 33) of Group D (*p* = 0.008). The first group showed a duration of stay less than 15 days in 75% (*n* = 15) of cases vs. 89% (*n* = 65) of the second (*p* = 0.001). Furthermore, the indirect impact of COVID-19 management could be seen, as there was higher mortality among patients who underwent a transfer before the surgical procedure, compared to other patients (25% vs. 6.85%; *p* = 0.02). Up to the third month of follow-up, the subjects of Group C attained worse clinical and radiographic outcomes than Group D (HHS 80–89 points: 10% vs. 37%; Rush Score 18–24 Rush Score: 5.20% vs. 30.5%) (*p* = 0.00001; *p* = 0.003). For purely cognitive purposes, it was found that 54.22% of patients in our study sample completed the vaccination cycle (double dose) anti- SARS-CoV-2 by June 2021, but none completed the cycle during the period of hospitalization.

## 4. Discussion

Cases of an unidentified form of viral pneumonia were first reported in Wuhan city, China in December 2019. The virus is believed to be acquired from a zoonotic source. This unknown virus gradually spread across the whole world. The common symptoms observed in patients with COVID-19 are fever, cough, severe headache, and fatigue. Italy was one of the worst-affected countries in the first months of the pandemic [5]. A series of containment policies have been implemented since the start of the outbreak. The Italian government declared the quarantine of 11 municipalities in Northern Italy on 21 February, which was then extended to the whole country the next day [6,10]. The restrictions adopted on 13 October 2020 implemented the containment of SARS-CoV-2 contagion [20].

The pandemic management led to a significant change in orthopedic clinical and surgical activity. During this historic period, the incidence of proximal femur fractures in patients over 65 years of age did not show a reduction in cases [6,27]. The Local Health Department, therefore, had to undertake some managerial choices to allow a reorganization of the hospitalized patients. The femur fracture in the fragile patient requires multidisciplinary treatment, an approach that is difficult to manage even in a non-pandemic time [11,14]. These fragile subjects, victims of trauma, must receive surgery urgently [11,27]. Numerous studies support the close correlation between increased mortality and delayed orthopedic treatment [27]. The metanalysis conducted by Moja et al. shows how a delay in surgery beyond 48 h increases not only the risk of mortality, but also the risk of prolonged hospitalization [23]. Simunovic’s study shows that the delay in treatment also leads to an increase in non-orthopedic perioperative complications [24]. As pointed out in some studies in recent years, the pandemic has greatly influenced the timing of femur fractures management, increasing time before diagnosis and treatment, thus increasing post-surgical mortalities [15,16,29]. The Coronavirus disease (COVID-19) has created severe humanitarian and socio-economic issues in the world [15,30].

The innovation of our study, compared to previous studies, was the analysis of the incidence of intra-hospital infection with SARS-CoV-2 in a sample consisting of patients with proximal femur fractures negative to molecular swab at the time of hospitalization. It also investigated how the pandemic management influenced the clinical and functional results of the patients under examination. On 21 December 2020 the European Medicine Agency (EMA) authorized the first vaccine against SARS-CoV-2, and on 27 December the first vaccination campaign against SARS-CoV-2 in Italy was launched, aimed at health staff and fragile populations. In view of these new events, it was decided to include in our analysis an even more up-to-date variable: the influence of the vaccination of health professionals compared to the incidence of infection in our inpatient population.

The proximal femur fractures included in our study (from 15 October 2020 to 31 March 2021) were 93, average age 83.75 years (65–98). The subjects included had to be necessarily negative to the PCR swab carried out in the E.R. The incidence of SARS-CoV-2 infection during the stay was 11.83% (11 patients), less than some data from the literature indicates [4,31]. It has been shown that 100% of the subjects infected were hospitalized at Time 0 (from 15 October 2020 to 10 February 2021), a period during which less than 80% of health care personnel were vaccinated against SARS-CoV-2 (*p* = 0.02). A long period spent in the E.R. before hospitalization led to an increased risk of onset of disease due to Coronavirus (time spent in E.R. > 24 h: 36% Group A vs. 22% Group B; *p* = 0.019). SARS-CoV-2 disease increased discharge times (stay > 15 days: Group A 27.27% vs. Group B 11.11%; *p* = 0.00001), intra-hospital mortality (Group A 18% vs. Group B 1.23%; *p* = 0.003) and mortality within 30 days after discharge (Group A 36% vs. Group B 6%; *p* = 0.007). Functional and radiographic outcomes were also lower in those who found the virus during the hospital stay (HHS Good: Group A 18.18% vs. Group B 33.33%; *p* = 0.00001). During the period examined, due to management problems related to the pandemic, it was necessary to transfer 20 victims of proximal femur fractures (21.51% of the sample analyzed) to reference hospitals. Ten percent of the transferred subjects underwent surgery within 24 h of E.R. access, compared to 22% of patients belonging to the other group (*p* = 0.00007). A stay of less than 15 days distinguished 89% of the subjects not transferred compared to 75% of the other patients (*p* = 0.001). The transfer of patients and therefore the delay of treatment negatively affected their prognosis: Mortality was higher in this group compared to those not transferred (25% Group C vs. 6.85% Group D; *p* = 0.02). Functional results were also better in patients admitted without transfer (HHS Good: 10% Group C vs. 37% Group D; *p* = 0.00001).

The limits of the study are many: first, the low sample size. The short-term follow-up does not allow us to have a complete picture of the outcomes. Moreover, the peculiarity of the health conditions examined does not allow to reproduce and compare the same analyses in other samples.

## 5. Conclusions

The impact of the pandemic from SARS-CoV-2 compared to the clinical course of patients treated for a proximal femur fracture was statistically significant. Patients with Coronavirus during hospitalization compared to negative patients, obtained poor short-term radiographic and functional results and increased peri-operative mortality. The incidence of intra-hospital infection was high over the period in which health professionals were not yet covered by the anti-COVID vaccination cycle. Patients who were transferred between two hospitals, due to pandemic-related management issues, also achieved reduced outcomes compared to non-transferred cases, with increased mortality. From our study, therefore, it appears that delayed treatment in fragile patients entails an increased risk of complications, reduced functional recovery, and increased mortality.

## Figures and Tables

**Table 1 jcm-11-06605-t001:** Demographic data of the sample.

	Total Sample
**Patients**	93
**Sex**	
Male	20 (21.50%)
Female	73 (78.5%)
**Age**	83.75 years (65–98)
**Home status**	
Own Home	78 (83.87%)
Nursing Home	15 (16.13%)
**Relevant comorbitidy**	
no comorbidity	15 (13.95%)
beetween 1 and 3	64 (59.52%)
>3	8 (7.44%)
**PPM classification**	
≤3 points	7 (6.51%)
beetween 4 and 5 points	44 (40.92%)
>5 points	34 (31.62%)
**NHFS score**	2.8-11.8 (5.24%)
**ASA**	
≤2 grade	25 (23.25%)
3 grade	44 (40.92%)
4 grade	12 (16%)
**OTA/AO**	
31A1	3 (3.23%)
31A2	42 (45.16%)
31A3	12 (12.91%)
31B/C	30 (31.26%)
**Surgical procedure**	
ORIF	52 (55.91%)
Partial hip replacement	19 (20.43%)
Total hip replacement	2 (2.15%)

**Table 2 jcm-11-06605-t002:** Positive cohort with the negative Cohort.

	COVID-19 Positive Cohort (Group A)	Covid-19 Negative Cohort (Group B)	*p* Value
	11 (11.83%)	82 (88.17%)	
**Hospitalization at Time 0**	11 (100%)	43 (52.43%)	*p* = 0.02
**Hospitalization at Time 1**	0	39 (47.56%)	*p* = 0.02
**Intensive Care Unit**	8 (72.73%)	4 (4.88%)	*p* = 0.000019
**E.R. > 24 h**	7 (63.64%)	18 (22%)	*p* = 0.019
**Length of stay, average**	21 days (10–32)	14 days (7–22)	*p* = 0.00001
**Mortality in ward**	2 (18.18%)	1 (1.22%)	*p* = 0.003
**Mortality from surgery to 3 months of follow-up**	4 (36%)	5 (6%)	*p* = 0.007
**HHS 80–89 points (3 months)**	2 (18.18%)	27 (33.33%)	*p* = 0.00001
**RUSH score 18–24 points (3 months)**	1 (10.20%)	33 (40.42%)	*p* = 0.00002

**Table 3 jcm-11-06605-t003:** Main results obtained comparing transferred patients’ cohort with the not transferred group.

	Transferred Patients’ Cohort (Group C)	Not Transferred Patients’ Cohort (Group D)	*p* Value
	20 (21.50%)	73 (78.50%)	
**Surgery < 24 h**	2 (10%)	18 (24.66%)	*p* = 0.00007
**Length of stay < 15 days**	15 (75%)	65 (89%)	*p* = 0.001
**Mortality from surgery to 3 months of follow- up**	5 (25%)	5 (6.85%)	*p* = 0.02
**HHS 80–89 points (3 months)**	2 (10%)	27 (37%)	*p* = 0.00001
**RUSH score 18–24 points (3 months)**	1 (5.20%)	22 (30.5%)	*p* = 0.003

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
