# Peer review of "Management of Femur Fractures during COVID-19 Pandemic Period: The Influence of Vaccination and Nosocomial COVID-19 Infection"

_jcm, 2022, doi:10.3390/jcm11226605_

Round 1
Reviewer 1 Report
The present manuscript analyzed a cohort of hip fractures during the COVID-19 pandemic, and compare results of the fracture analyzing COVID-19 infection in patients during the stay, percentage of COVID-19 vaccination of health professionals and if the patients have to be transferred before the surgery. It is an interesting topic.
Methods
Could you explain why have you excluded patients diagnosed with COVID-19 at the entrance in the Emergency Department? It could be interesting to know the results of this cohort of patients. If you have this data, could be interesting to add it to the paper.
3 months of follow-up seems to be a short period to evaluate results. It would be interesting to have the results with a 6-months of follow-up, and ideally also the results after a year
Results
Most of the data from the first paragraph of the results are also repeated in Table I. Repeated data could be removed. Please, also revise repeated data in the text and tables II al III.
Table I: It would be interesting to have, in addition to the total data, the data stratified by groups: Covid+/covid-; Transferred/not tranfered.
In the 2nd paragraphs of “results” “3.1. Covid-19 positive cohort vs Covid-19 negative cohort” authors wrote “Comparing demographics characteristics of the Group A 189 to the Group B (Covid-19 negative cohort), the average age (p= 0.31), the gender (p= 0.41), 190 the ASA score (p= 0.40), the PPM Score (p= 0.38), the NHFS (p= 1.22) and the type of fractures 191 were comparable (p= 0.10) (Table II)”. I have not been able to find this data in table II
In the 3rd paragraphs of “results” “3.2. Transferred patients’ cohort vs not transferred patients’ cohort” authors wrote “ Comparing Group C vs D (not transferred patients), the average age (p= 209 0.61), the gender (p= 0.71), the ASA score (p= 1.40), the PPM Score (p= 0.22), the NHFS (p= 210 0.45) and the type of fractures were comparable (p= 2.10) (Table III). I have not been able to find this data in table III.
It would be interesting to describe how many of the transferred/not transferred patients had positive COVID test.
Author Response
Methods
Could you explain why have you excluded patients diagnosed with COVID-19 at the entrance in the Emergency Department? It could be interesting to know the results of this cohort of patients. If you have this data, could be interesting to add it to the paper.
The patients diagnosed with COVID-19 at the entrance were transferred to another hospital of reference, according to the regional rules. We have not the data of that sample.
3 months of follow-up seems to be a short period to evaluate results. It would be interesting to have the results with a 6-months of follow-up, and ideally also the results after a year
The study ended with this follow up time, we are archiving data related to a longer follow up for another research.
Results
Most of the data from the first paragraph of the results are also repeated in Table I. Repeated data could be removed. Please, also revise repeated data in the text and tables II al III.
We prefer to leave the data in the text completely and have an immediate graphic reference with the same data within the table. If our methodology is not adequate, we will have to eliminate the tables.
Table I: It would be interesting to have, in addition to the total data, the data stratified by groups: Covid+/covid-; Transferred/not tranfered.
We add the data stratified by groups (see the attachment).
In the 2nd paragraphs of “results” “3.1. Covid-19 positive cohort vs Covid-19 negative cohort” authors wrote “Comparing demographics characteristics of the Group A 189 to the Group B (Covid-19 negative cohort), the average age (p= 0.31), the gender (p= 0.41), 190 the ASA score (p= 0.40), the PPM Score (p= 0.38), the NHFS (p= 1.22) and the type of fractures 191 were comparable (p= 0.10) (Table II)”. I have not been able to find this data in table II.
In the 3rd paragraphs of “results” “3.2. Transferred patients’ cohort vs not transferred patients’ cohort” authors wrote “Comparing Group C vs D (not transferred patients), the average age (p= 209 0.61), the gender (p= 0.71), the ASA score (p= 1.40), the PPM Score (p= 0.22), the NHFS (p= 210 0.45) and the type of fractures were comparable (p= 2.10) (Table III). I have not been able to find this data in table III.
There was an error in describing the tables. The general data table was not included in the text. I think it is necessary to remove the reference to the tables in that paragraph and add it at the end of paragraph 3.1 and 3.2, respectively.
It would be interesting to describe how many of the transferred/not transferred patients had positive COVID test.
The percentage of positive COVID In the transferred patient’s group was 18.18%. This data was not significant and it was not included in the manuscript.

Reviewer 2 Report
Well written manuscript that reflects management of femur fractures during COVID-19.
The manuscript can be strengthened by incorporating the following points:
1. Title: include the type of study and the country in which the study was conducted.
2. Keywords: Do not include the words that are already stated in the title.
3. Introduction and discussion: Should incorporate the following as in the presented manuscript it appears very concise:
- Include a short note on the origin of COVID-19 (refer and cite: doi: 10.1136/postgradmedj-2020-138234
- Compare the COVID-19 states in the country of your study with other regions (refer and cite: doi: 10.3389/fpubh.2022.844333)
- The role of vaccination status in relation to management of COVID-19 patients (refer and cite: doi: 10.3390/vaccines9101064.)
- How the government and health care has imbibed regulations to combat COVID-19 targeting its predictors (refer and cite: doi: 10.1136/postgradmedj-2021-141365)
- The brief note on general treatment of COVID-19 and its implications on the vaccine hesitancy (refer and cite: doi: 10.3389/fphar.2022.742273.
- Include other forms of representations in your figures (eg. Bar charts, histograms), color the images for better viewership.
Statistical evaluation, data collection and results:
1. Mention if missing cases were encountered during data collection and how were they eliminated during statistical evaluation.
2. Indicate the dependent and independent variables in your study.
3. Why was Student’ T-test used?
4. Was there any bias generated and how was it contained to bring about a favorable statistically significant index?
5. Provide a sample of data collection sheet as a supplementary file.
Author Response
- Title: include the type of study and the country in which the study was conducted.
Management of femur fractures during Covid-19 pandemic period: the influence of vaccination and nosocomial Covid-19 infection, a prospective Italian study
- Keywords: Do not include the words that are already stated in the title.
We can add some keywords like “Coronavirus; hip fracture, neck femur fracture, lockdown, SARS-CoV-2, trauma, geriatric patients”.
- Introduction and discussion: Should incorporate the following as in the presented manuscript it appears very concise:
- Include a short note on the origin of COVID-19 (refer and cite: doi: 10.1136/postgradmedj-2020-138234
We have added a note about the origin of COVID-19. Please see the attachment (in red the added note).
- Compare the COVID-19 states in the country of your study with other regions (refer and cite: doi: 10.3389/fpubh.2022.844333)
Please see the attachment (in red the added note).
- The role of vaccination status in relation to management of COVID-19 patients (refer and cite: doi: 10.3390/vaccines9101064.)
We have added a note on this in the text (in red the added note).
- How the government and health care has imbibed regulations to combat COVID-19 targeting its predictors (refer and cite: doi: 10.1136/postgradmedj-2021-141365)
Please see the attachment (in red the added note).
- The brief note on general treatment of COVID-19 and its implications on the vaccine hesitancy (refer and cite: doi: 10.3389/fphar.2022.742273.
Please see the attachment (in red the added note).
- Include other forms of representations in your figures (eg. Bar charts, histograms), color the images for better viewership.
Please see the attachment.
Statistical evaluation, data collection and results:
- Mention if missing cases were encountered during data collection and how were they eliminated during statistical evaluation.
We did not have patients lost to follow up, probably due to the short period analyzed. All the unclear and incomplete data have been eliminated during statistical evaluation and the results corrected according to data not available.
Indicate the dependent and independent variables in your study.
We considered independent variables: sex, age, residence, the type of fracture, the comorbidity, Sars Cov-2 infection, state of vaccination against SARS Cov2 of health workers and inpatient, delay in the days following the presentation to the Emergency Department compared to the surgery and the beginning of physiotherapy, number of transfers.
We considered dependent variables: ASA classification, the PPM Score, the NHFS, the type of surgical procedure performed, oxygen therapy during the stay, the post-surgical complications, the ROM and the HHS.
- Why was Student’ T-test used?
The Student’s Test T for continuous data were utilized to assess differences difference between independent sample for example comparing the age, the period between the E.R. and the surgery, the time between the surgery and the rehabilitation, the surgical time, the length of stay. As indicated in the manuscript, only statistically significant results for values of p<0,05 were considered relevant.
- Was there any bias generated and how was it contained to bring about a favorable statistically significant index?
As indicated in the study limits, the main expected bias could be related to the small sample size. To avoid this bias, the study design was drawn up prospectively with very precise exclusion criteria to ensure a well-defined target population.
- Provide a sample of data collection sheet as a supplementary file.
Please see the attachment. The program does not allow to load excel files. We have inserted a demonstration of the data collection at the end of the manuscript. The image is not to be added to the text.

Round 2
Reviewer 2 Report
The authors have incorporated the suggested comments in their revised manuscript.